# FKBP4 regulates 5-fluorouracil sensitivity in colon cancer by controlling mitochondrial respiration

Zhenyu Zhu[1],*, Qingsheng Hou[1],*, Bishi Wang[1], Changhao Li[1], Luguang Liu[1], Weipeng Gong[1], Jie Chai[2], Hongliang Guo[1], Yanhan Jia[3]

**Mitochondrial respiration and metabolism play a key role in the pathogenesis and progression of colon adenocarcinoma (COAD). Here, we report a functional pool of FKBP4, a co-chaperone protein, in the mitochondrial intermembrane space (IMS) of colon cancer cells. We found that IMS-localized FKBP4 is essential for the maintenance of mitochondrial respiration, thus contributing to the sensitivity of COAD cells to 5-fluorouracil (5-FU). Mechanistically, FKBP4 interacts with COA6 and controls the assembly of the mitochondrial COA6/SCO1/SCO2 complex, thereby governing COA6-regulated biogenesis and activity of mitochondrial cytochrome c oxidase (complex IV). Thus, our data reveal IMS-localized FKBP4 as a novel regulator of 5-FU sensitivity in COAD, linking mitochondrial respiration to 5-FU sensitivity in COAD.**

## Introduction

Colon adenocarcinoma (COAD) is one of the most common and lethal malignant tumors of the digestive system (Siegel et al, 2020). As a first-line strategy for COAD treatment, 5-fluorouracil (5-FU)–based chemotherapy shows impressive clinical benefits. However, because of intrinsic insensitivity or acquired resistance to 5-FU treatment, the overall response rate to 5-FU in COAD is limited (Blondy et al, 2020; Ghafouri-Fard et al, 2021). Thus, identification of the biological mechanisms governing sensitivity of COAD cells to 5-FU would be helpful to improve the therapeutic response and prolong survival in COAD patients.

Metabolic reprogramming plays essential roles in cancer malignancy and progression (Wallace, 2012; Willems et al, 2015). In comparison to other types of human cancer, COAD cells exhibit more mitochondrial metabolism (Kaldma et al, 2014; Chekulayev et al, 2015; Sun et al, 2018). It has been widely shown that elevated mitochondrial respiration promotes colon cancer cell proliferation and contributes to colon cancer tumorigenesis and progression

(Boyle et al, 2018; Lin et al, 2018; Sun et al, 2018; Wen et al, 2019). Besides, altered mitochondrial respiration has been shown to sensitize colorectal cancer cells to chemotherapy, including 5-FU (Liang et al, 2008; Denise et al, 2015; Vellinga et al, 2015; Bosc et al, 2017; Yun et al, 2019). However, little is known about the molecular mechanisms and key regulators of mitochondrial respiration that confer 5-FU sensitivity in COAD cells.

Immunophilin protein FKBP4 (also known as FKBP52) is an HSP90-associated co-chaperone that controls assembly of various protein complexes (Martinez et al, 2013; Zong et al, 2021). FKBP4 acts as a scaffold protein to facilitate the interactions between key components of several cancer-promoting signaling pathways (Mange et al, 2019; Zong et al, 2021). High expression of FKBP4 is associated with the malignancy and progression of several types of human cancer (Rees-Unwin et al, 2007; Bhowal et al, 2017; Federer-Gsponer et al, 2018; Mange et al, 2019; Liu & Gao, 2021; Zong et al, 2021). However, no study has yet addressed the role of FKBP4 in regulating sensitivity of COAD cells to 5-FU treatment. The subcellular localization profiles of FKBP4 based on The Human Protein Atlas database show both cytoplasmic and nuclear distribution of FKBP4 in a variety of cell types (Thul et al, 2017). Nevertheless, the compartmentalized activity and function of FKBP4 still remains unclear.

COA6 is an assembly factor that controls stability and activity of mitochondrial cytochrome c oxidase (complex IV) (Maghool et al, 2020; Swaminathan & Gohil, 2022). COA6 cooperates with SCO1 and SCO2 in a complex to manipulate copper delivery pathway, thereby dictating the copper-dependent biogenesis of complex IV subunits and modulating mitochondrial respiration (Pacheu-Grau et al, 2015, 2020; Stroud et al, 2015; Soma et al, 2019). Loss-of-function COA6 mutant leads to complex IV deficiency and thus causes human mitochondrial disease (Ghosh et al, 2014; Baertling et al, 2015; Pacheu-Grau et al, 2015). However, the role of COA6-controlled mitochondrial respiration in COAD, a mitochondria-dependent disease, has not yet been addressed.

In this study, we uncover compartmentalized distribution of FKBP4 in the cytoplasm, nuclei, mitochondrial intermembrane

---

[1]Gastrointestinal Surgery Ward II, Shandong Cancer Hospital and Institute, Shandong First Medical University and Shandong Academy of Medical Sciences, Jinan, China [2]Gastrointestinal Surgery Ward I, Shandong Cancer Hospital and Institute, Shandong First Medical University and Shandong Academy of Medical Sciences, Jinan, China [3]Sichuan Cancer Hospital and Institute, Sichuan Cancer Center, School of Medicine, University of Electronic Science and Technology of China, Chengdu, China

Correspondence: sdguohongliang@gmail.com; jiayanhan@outlook.com
*Zhenyu Zhu and Qingsheng Hou contributed equally to this work and share first authorship.

space (IMS), and mitochondrial matrix (MM) of COAD cells. We found that IMS-localized FKBP4 plays an essential role in the regulation of mitochondrial respiration and thus confers sensitivity of COAD cells to 5-FU. Mechanistically, FKBP4 manipulates the establishment of the COA6/SCO1/SCO2 complex, thereby facilitating the COA6-mediated complex IV biogenesis and activity. Importantly, up-regulated FKBP4 and COA6 in 5-FU-treated COAD patients are correlated with poor prognosis. Thus, our analyses define IMS-localized FKBP4 as a novel regulator of 5-FU sensitivity in COAD, providing a molecular link between altered mitochondrial respiration and 5-FU sensitivity in COAD.

# Results

## FKBP4 is up-regulated in colon cancer cells and localized in both mitochondrial intermembrane space and matrix

We first analyzed the gene expression profile of FKBP4 in non-malignant colon and COAD specimens from The Cancer Genome Atlas (TCGA) and observed a significant up-regulation of FKBP4 in COAD samples in comparison to the normal colon tissues (Fig 1A). Accordingly, immunoblotting analysis of the FKBP4 protein level in various human COAD and noncancerous colonic epithelial cell lines showed that all the tested COAD cell lines expressed higher levels of FKBP4 in comparison to noncancerous colonic epithelial cells (Fig 1B). Next, we analyzed the subcellular localization of FKBP4 in COAD cells using the cell fractionation strategy. In line with what is shown in the Human Protein Atlas (Thul et al, 2017), we observed that FKBP4 was expressed in the cytoplasm, nuclei, and mito-chondria of both HCT116 and SW480 cells (Fig 1C). Further investigation of submitochondrial localization of FKBP4 by proteinase K protection assay revealed that FKBP4 was located in both IMS and MM (Fig 1D).

We next attempted to elucidate the mechanism by which FKBP4 is imported into mitochondria. The tetratricopeptide repeat (TPR) domain of FKBP5, another member of the FKBP family sharing 60% identity and 75% similarity with FKBP4, was shown to be responsible for its mitochondrial localization via interaction with HSP90/HSP70 (Gallo et al, 2011). Thus, we wondered whether the same mechanism confers the mitochondrial translocation of FKBP4. To address this question, we established HCT116 cells bearing the K354A mutation in the TPR domain of FKBP4 (FKBP4-K354A) (Fig S1A) that has been widely shown to abolish FKBP4/HSP90 interaction (Cheung-Flynn et al, 2003; Riggs et al, 2003; Zong et al, 2021). In accordance with these previous studies, we observed that FKBP4/HSP90 and FKBP4/HSP70 interaction were abrogated by FKBP4-K354A mutation (Fig S1B). However, the mitochondrial localization of FKBP4 was not affected by the FKBP4-K354A mutation (Fig S1C), suggesting that mitochondrial localization of FKBP4 was not controlled by its TPR domain–mediated interaction with HSP90/HSP70.

Proteins residing in mitochondria often contain mitochondrial-targeting sequences (MTSs) within their N-terminal amino acid sequence (Wiedemann & Pfanner, 2017; Hansen & Herrmann, 2019). We thus investigated the amino acid sequence of FKBP4 for identification of its putative MTS sequence. However, we could not observe a canonical MTS sequence in FKBP4 by online MTS prediction tools (Claros & Vincens, 1996; Fukasawa et al, 2015), suggesting that FKBP4 contains an unconventional targeting sequence that is not recognized by the current computational prediction or is imported into mitochondria by another unknown mechanism. Both possibilities have been previously shown (Marchenko et al, 2000; Lee et al, 2005; Li et al, 2010; Chatterjee et al, 2016; Wiedemann & Pfanner, 2017).

Taken together, these results demonstrate that, in addition to the nucleus and cytoplasm, FKBP4 resides in mitochondria, more precisely, in both IMS and MM.

## Loss of FKBP4 reduces mitochondrial respiration in COAD cells

To evaluate the functional impact of FKBP4 loss on mitochondria in COAD cells, we first generated the FKBP4 depleted HCT116 and SW480 cells using siRNA-mediated knockdown strategy. Next, we investigated the mitochondrial respiration profiles of control (siCtrl) and FKBP4-depleted (siFKBP4) COAD cells by performing Seahorse Cell Mito Stress Test analysis. When comparing the respiration profiles of siCtrl with siFKBP4 cells, we observed significantly reduced basal, ATP-linked respiration and maximal respiratory capacity in the knockdown cells (Fig 1E–J). To assess whether impaired mitochondrial respiration upon FKBP4 depletion leads to compensatory alterations in glycolysis, we monitored the glycolytic levels of siCtrl and siFKBP4 cells using Seahorse XF Glycolysis Stress Test analysis. Of note, loss of FKBP4 did not result in any changes in glycolysis or glycolytic capacity of COAD cells (Fig S1D–G). These data suggest that depletion of FKBP4 reduces mitochondrial respiration in COAD cells.

## IMS-localized FKBP4 is indispensable for maintaining mitochondrial respiration in COAD cells

To dissect the contribution of compartmentalized FKBP4 more concretely in COAD mitochondrial respiration, we generated various 6xHis-tagged FKBP4-derivative COAD cells lines (Figs S2A and S3A). In particular, we enforced FKBP4 transport to IMS by fusing a well-characterized MTS sequence of SMAC (referred to as MTS1 from here on) at its N-terminus (Newman et al, 2016; Saita et al, 2017). Similarly, import of FKBP4 into MM was imposed by fusing the MTS sequence of COX4A (referred to as MTS2 from here on) at its N-terminus (Chatterjee et al, 2016). Nuclear transport of FKBP4 was implemented by fusing the nuclear localization sequence of c-Myc (referred to as NLS from here on) at its N-terminus (Dang & Lee, 1988; Chiu et al, 2018). We next investigated whether impaired mitochondrial respiration upon FKBP4 depletion could be rescued by ectopic expression of MTS1-FKBP4, MTS2-FKBP4, and NLS-FKBP4. For this purpose, we infected COAD cells with lentivirus driving expression of either the empty vector (EV) control or one of the FKBP4 derivatives followed by treating these cells with non-targeting negative control siRNA (siCtrl) or siRNA that targets the 3′ UTR of the endogenously transcribed FKBP4 mRNA (siFKBP4_3′UTR, Figs S2A and S3A). Cell fractionation and the subsequent purity assessment by Western blot analysis with anti-FKBP4 and anti-6xHis antibodies validated the exclusive mitochondrial

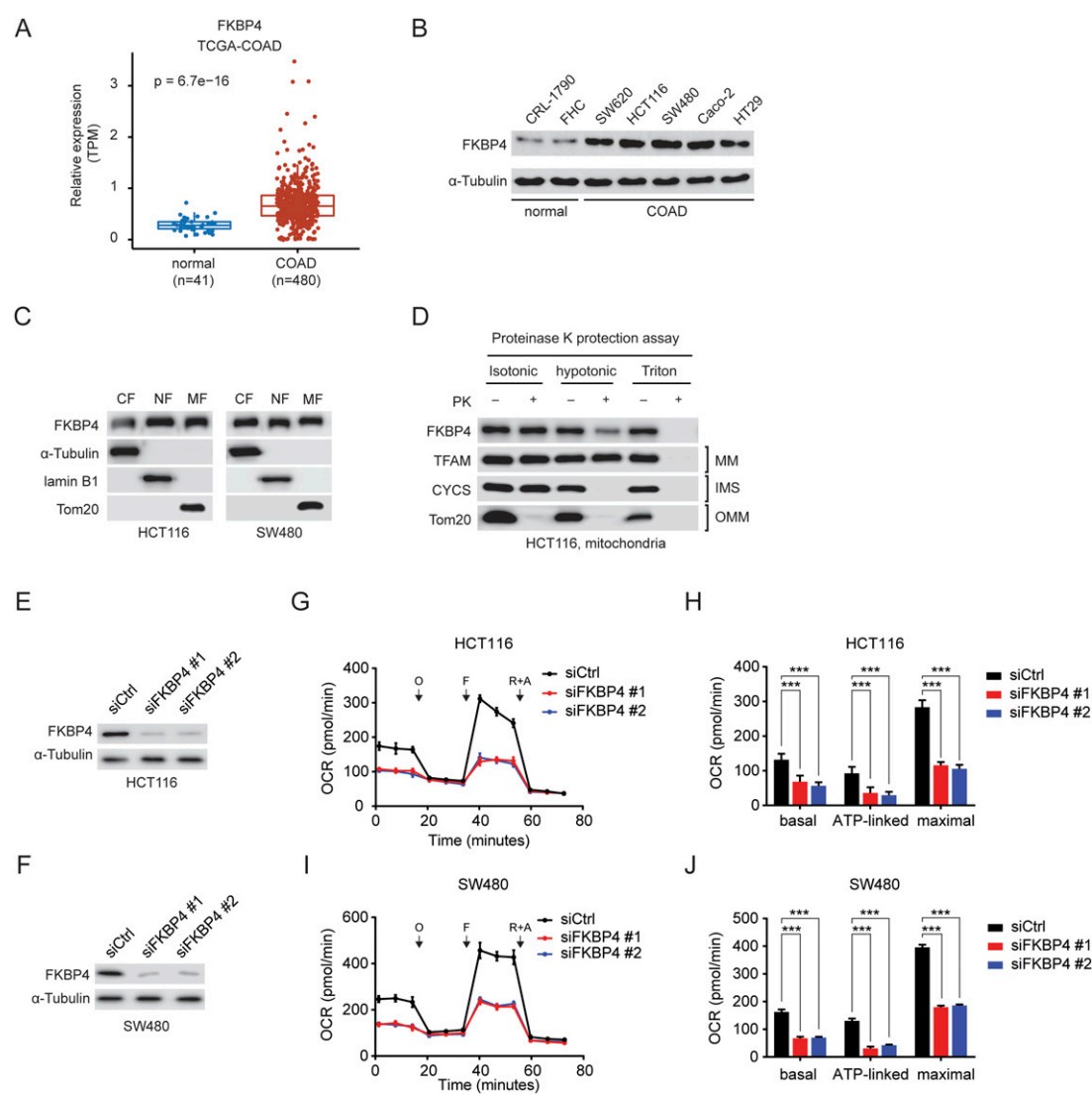

**Figure 1. FKBP4 shows mitochondrial localization and is required for the maintenance of mitochondrial respiration in CRC cells.**
**(A)** Box plot of the FKBP4 expression level in normal (n = 41) and colon tumor (n = 480) samples based on The Cancer Genome Atlas (TCGA) dataset (TCGA-COAD). The *P*-value of the Wilcoxon rank-sum test is shown in the plot. **(B)** Western blot analysis of FKBP4 expression in various types of noncancerous colonic epithelial cell lines and COAD cell lines. **(C)** Western blot analysis of the indicated proteins in cytoplasmic (CF), nuclear (NF), and mitochondrial fractions (MF) of HCT116 and SW480 cells. **(D)** Proteinase K protection assay of the mitochondria freshly isolated from HCT116 cells. PK, proteinase K; MM, mitochondrial matrix; IMS, intermembrane space; OMM, outer mitochondrial membrane. Well-characterized mitochondrial proteins, TFAM (MM protein), CYCS (IMS protein), and Tom20 (OMM protein) were used as the control, indicating the extent of digestion. **(E, F)** Western blot analysis of FKBP4 expression in HCT116 (E) and SW480 (F) cells treated with siCtrl or two different siRNAs targeting FKBP4. **(G, H, I, J)** Profiles of mitochondrial respiration over time in HCT116 (G, H) and SW480 (I, J) treated with siCtrl or two different siFKBP4 using the Seahorse XFe 96 analyzer. O, oligomycin; F, carbonyl cyanide-4-(trifluoromethoxy)phenylhydrazone (FCCP); R+A, rotenone and antimycin A. **(H, J)** Basal and ATP-linked respiration as well as maximal respiratory capacity were calculated. \*\*\**P* < 0.001 by the two-tailed *t* test. Data are presented as mean + SD. n = 6.

localization of ectopically expressed MTS1-FKBP4 and MTS2-FKBP4 as well as exclusive nuclear localization of NLS-FKBP4 (Figs 2A and E and S3B). Importantly, siFKBP4_3′UTR specifically and effectively depleted only endogenous FKBP4 without affecting the ectopically expressed FKBP4-derivatives (Figs 2A and E and S3B). We further performed proteinase K protection assay in COAD cells treated with siFKBP4_3′UTR in combination with MTS1-FKBP4 or MTS2-FKBP4. The results showed that ectopically expressed MTS1-FKBP4 was imported into IMS exclusively (Fig 2B). In contrast, MTS2-FKBP4 showed only MM localization (Fig 2F). Furthermore, respiration

profile analysis showed that mitochondrial respiration of EV transfected control cells was strongly impaired upon loss of endogenous FKBP4, whereas the respiration of MTS1-FKBP4 ectopically expressed COAD cells was not affected by depleting endogenous FKBP4 (Figs 2C and D and S2B and C). In contrast, MTS2-FKBP4 (Figs 2G and H and S2D and E) and NLS-FKBP4 (Fig S3C–F) failed to rescue the respiration defect caused by siFKBP4_3′UTR. Taken together, these data provide strong evidence for a functional significance of FKBP4 in IMS and its indispensable role in the regulation of mitochondrial respiration.

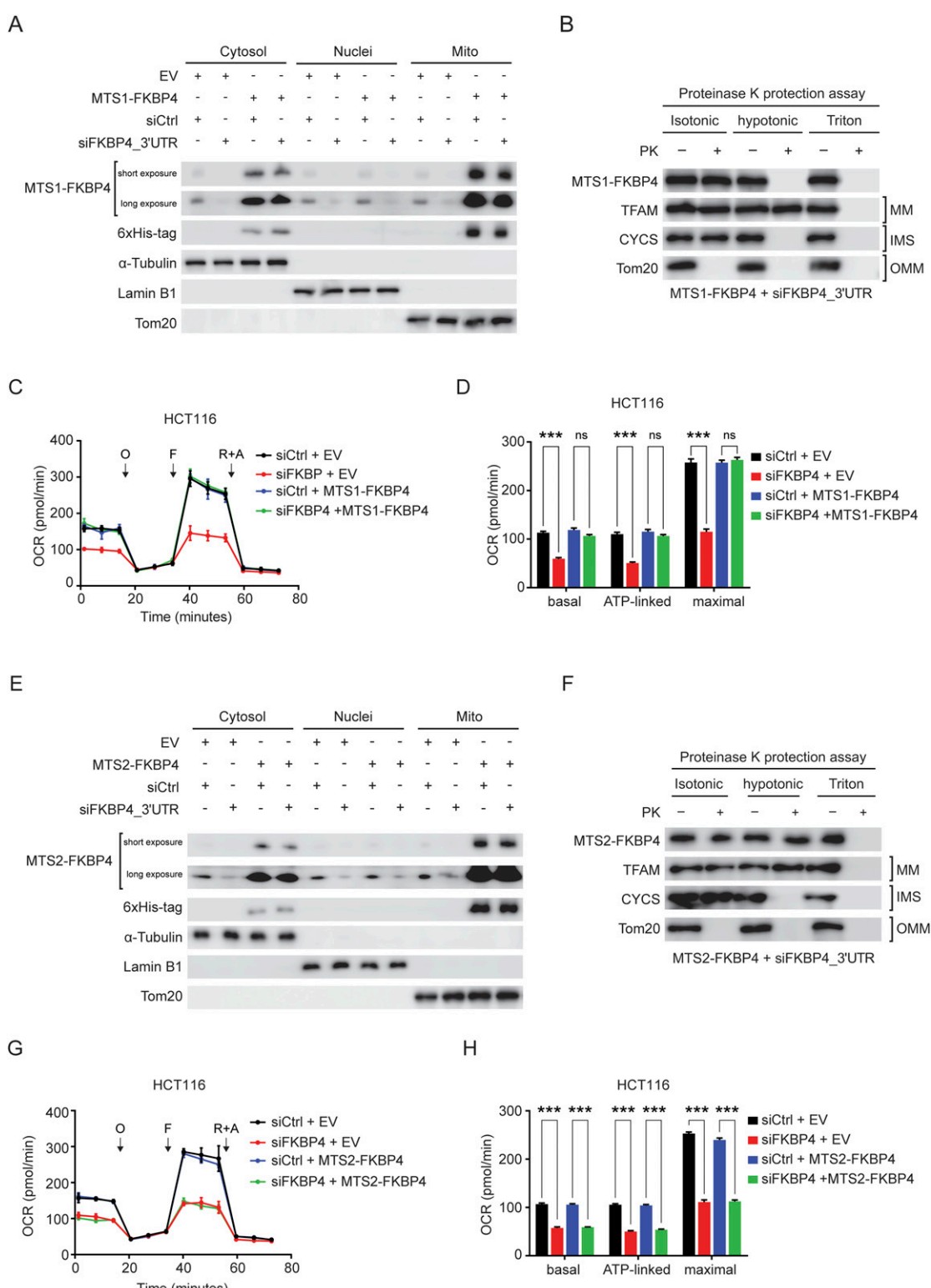

**Figure 2. IMS-localized FKBP4 plays a critical role in controlling mitochondrial respiration in COAD cells.**
**(A)** Immunoblotting for the indicated proteins in cytosolic, nuclear, and mitochondrial (Mito) fractions of HCT116 cells transfected with an empty vector (EV) or 6xHis-tagged FKBP4 fused with mitochondrial-targeting signal from SMAC (MTS1-FKBP4) followed by treatment with siCtrl or siRNA targeting 3' UTR of the endogenously transcribed FKBP4 mRNA (siFKBP4_3'UTR). **(B)** Proteinase K protection assay of mitochondria isolated from HCT116 cells that were transduced with MTS1-FKBP4 and transfected with siFKBP4_3'UTR. PK, proteinase K. MM, mitochondrial matrix; IMS, intermembrane space; OMM, outer mitochondrial membrane. TFAM (MM protein), CYCS (IMS protein), and Tom20 (OMM protein) were used as the control showing the extent of digestion. **(C, D)** Profiles of mitochondrial respiration over time in HCT116 cells

## Loss of IMS-localized FKBP4 confers enhanced sensitivity of COAD cells to 5-FU

We next examined whether the FKBP4 loss-mediated mitochondrial defect impedes proliferation of COAD cells. To this end, we performed sulforhodamine B (SRB) assay to determine the cell viability of COAD cells treated with siCtrl or siFKBP4. However, we did not observe proliferative inhibition in COAD cells upon FKBP4 knockdown (Fig S4A and B). Accordingly, loss of FKBP4 did not trigger apoptosis of COAD cells either (Fig S4C). Because the high FKBP4 level is known to promote migration and invasion of lung cancer cell (Meng et al, 2020; Zong et al, 2021), we thus wondered whether it plays a similar role in COAD cells. However, we did not observe a significant change in the migration/invasion level upon FKBP4 depletion in COAD cells (Fig S4D and E), suggesting that FKBP4 does not play a role in controlling COAD cell migration and invasion.

Elevated mitochondrial respiration has been associated with 5-FU resistance in COAD cells (Denise et al, 2015; Vellinga et al, 2015; Bosc et al, 2017; Yun et al, 2019). We thus assessed the impact of FKBP4 loss on sensitivity of COAD cells to 5-FU and found that the IC50 values of 5-FU in both HCT116 and SW480 cells were dramatically decreased upon FKBP4 depletion, suggesting that loss of FKBP4 increased the sensitivity of COAD cells to 5-FU (Fig 3A and B). Importantly, the increased 5-FU sensitivity upon FKBP4 loss was successfully reversed by ectopically expressing IMS-localized MTS1-FKBP4 (Fig 3C and D). Similar to what we observed with mitochondrial respiration analysis, MM-localized MTS2-FKBP4 or nucleus-localized NLS-FKBP4 had no effect on 5-FU sensitivity (Fig 3E–H). Thus, the IMS pool of FKBP4 plays a critical role in modulating sensitivity of COAD cells to 5-FU. Loss of IMS-localized FKBP4 enhances 5-FU sensitivity in COAD cells.

## FKBP4 is indispensable for COA6-mediated biogenesis and activity of complex IV

We next gained insight into the molecular mechanism by which IMS-localized FKBP4 regulates mitochondrial respiration and 5-FU sensitivity. COA6 was shown to govern biogenesis and activity of mitochondrial respiratory chain complexes IV (also called cytochrome c oxidase), thus playing a key role in maintaining mitochondrial respiration (Maghool et al, 2020; Swaminathan & Gohil, 2022). By co-immunoprecipitation (co-IP) analysis, we observed that FKBP4 interacted with COA6 endogenously in the mitochondria of COAD cells (Fig 4A). Besides, FKBP4 also interacted with SCO1 and SCO2 (Fig 4A), which were found in a complex with COA6 and required for the role of COA6 in the regulation of complex IV assembly

(Pacheu-Grau et al, 2015, 2020; Stroud et al, 2015; Soma et al, 2019). Furthermore, FKBP4/COA6 interaction was confirmed to be direct using GST pull-down analysis (Fig 4B). We next asked whether FKBP4 depletion led to changes in the expression of COA6, SCO1, and SCO2. To this end, we performed immunoblotting analysis to determine the expression of these proteins in COAD cells upon treatment with siCtrl or siFKBP4. However, we observed that FKBP4 depletion did not affect the protein levels of COA6, SCO1, and SCO2 in the mitochondria of COAD cells (Fig 4C). Intriguingly, FKBP4 knockdown led to general down-regulation of the key subunits of complex IV, COX1, COX2, and COX3 (Fig 4C), indicating a defect in complex IV biogenesis. Importantly, the decreased levels of COXs were rescued by overexpression of MTS1-FKBP4 but not MTS2-FKBP4 or NLS-FKBP4 (Fig 4D and E), suggesting that IMS-localized (instead of the MM-localized or nuclear) FKBP4 is indispensable for sustaining complex IV biogenesis. Because the COA6/SCO1/SCO2 complex has been well-characterized as a key regulator of complex IV biogenesis by dictating the translation of mtDNA-encoded COXs (Pacheu-Grau et al, 2015, 2020; Stroud et al, 2015), we thus assumed that FKBP4 depletion negatively affected the function of the COA6/SCO1/SCO2 complex for controlling complex IV biogenesis. In support of our hypothesis, we observed that FKBP4 knockdown disrupted the interaction between COA6, SCO1, and SCO2 (Fig 4F), though the complex rupture did not lead to changes in submitochondrial localization of these three proteins (Fig 4G). Taken together, these data demonstrate that FKBP4 is indispensable for the assembly of the COA6/SCO1/SCO2 complex and thus plays an essential role in the regulation of COA6-mediated complex IV biogenesis and assembly.

Next, we examined the impact of FKBP4 loss on complex IV activity. The activity of complex IV was determined with a commercial ELISA-based assay kit in siCtrl, siFKBP4, or siCOA6 COAD cells. We observed that loss of FKBP4 dramatically reduced complex IV activity in both HCT116 and SW480 cells (Fig 4H and I). In accordance with the well-characterized role of COA6 in controlling complex IV biogenesis and activity, we also found that COA6 knockdown caused significant loss of complex IV activity (Fig 4H and I). Importantly, ectopic expression of MTS1-FKBP4 successfully rescued the impaired complex IV activity caused by endogenous FKBP4 loss but failed to rescue the COA6 depletion-triggered inhibition (Fig 4H and I). In contrast, ectopic expression of COA6 successfully rescued siCOA6-induced complex IV deficiency but failed to rescue the siFKBP4-caused defect (Fig 4H and I). Taken together, these findings indicate an essential role of FKBP4 in controlling the establishment of the COA6/SCO1/SCO2 complex, thereby modulating the activity of mitochondrial complex IV.

infected with an empty vector (EV) or MTS1-FKBP4 followed by the transfection of siCtrl or siRNA targeting 3′UTR of endogenous FKBP4 mRNA (siFKBP4). O, oligomycin; F, carbonyl cyanide-4-(trifluoromethoxy)phenylhydrazone (FCCP); R+A, rotenone and antimycin A. **(D)** Basal, ATP-linked respiration, and maximal respiratory capacity were calculated. ***P < 0.001; ns, not significant by the two-tailed t test. Data are presented as mean + SD. n = 6. **(E)** Western blot analysis of the indicated proteins in cytosolic, nuclear, and mitochondrial (Mito) fractions of HCT116 cells that were infected with an empty vector (EV) or 6xHis-tagged FKBP4 fused with mitochondrial targeting signal from COX4A (MTS2-FKBP4) followed by treatment with siCtrl or siFKBP4_3′UTR. **(F)** Proteinase K protection assay of mitochondria isolated from HCT116 cells that was transfected with MTS2-FKBP4 and treated with siFKBP4_3′UTR. PK, proteinase K; MM, mitochondrial matrix; IMS, intermembrane space; OMM, outer mitochondrial membrane. TFAM (MM protein), CYCS (IMS protein), and Tom20 (OMM protein) were used as the control showing the extent of digestion. **(G, H)** Mitochondrial respiration profiles of HCT116 cells transfected with an empty vector (EV) or MTS2-FKBP4 in combination with the treatment of siCtrl or siRNA targeting 3′UTR of endogenous FKBP4 mRNA (siFKBP4). O, oligomycin; F, carbonyl cyanide-4-(trifluoromethoxy)phenylhydrazone (FCCP); R+A, rotenone and antimycin A. **(H)** Basal, ATP-linked respiration, and maximal respiratory capacity were calculated. ***P < 0.001 by the two-tailed t test. Data are presented as mean + SD. n = 6.

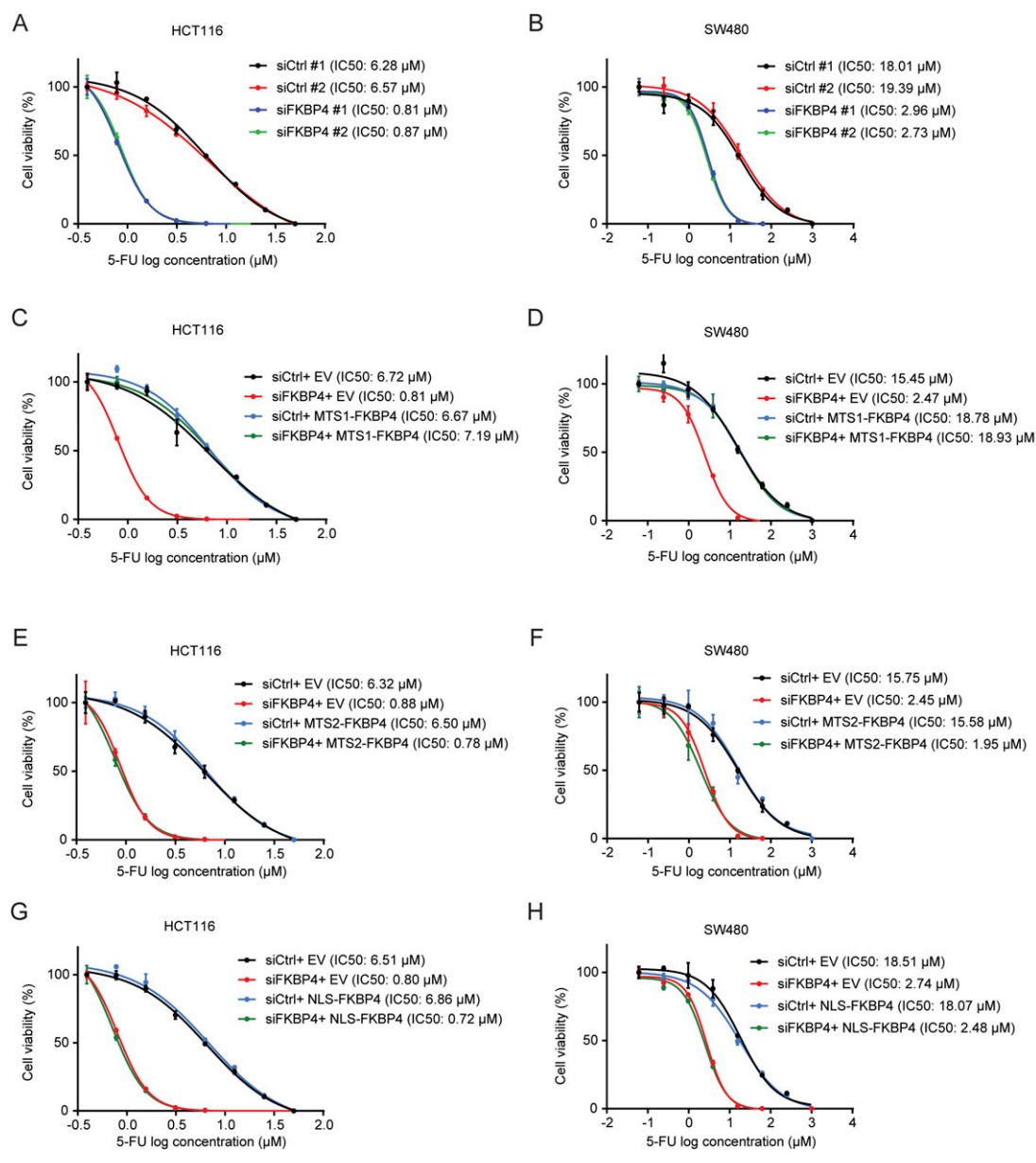

**Figure 3. IMS-localized FKBP4 regulates sensitivity of COAD cells to 5-FU.**
**(A, B)** Representative dose–response curves of 5-FU treatment in HCT116 (A) and SW480 (B) cells transfected with two different non-targeting control or FKBP4-targeting siRNAs. **(C, D, E, F, G, H)** Representative dose–response curves of 5-FU treatment in HCT116 (C, E, G) and SW480 (D, F, H) cells transfected with an empty vector (EV), MTS1-FKBP4 (C, D), MTS2-FKBP4 (E, F), or NLS-FKBP4 (G, H) in combination with siCtrl or siRNA targeting 3′ UTR of endogenous FKBP4 mRNA. **(A, B, C, D, E, F, G, H)** Twenty-four hours after siRNA transfection, cells were seeded into 96-well plate. After 24 h, cells were treated with different doses of 5-FU for 72 h, followed by cell viability analysis using sulforhodamine B (SRB) assay. IC50 values of 5-FU were indicated. Data are presented as mean ± SD. n = 3.

## COA6 loss impedes mitochondrial respiration and confers 5-FU sensitivity in COAD cells

Complex IV activity is essential to mitochondrial respiration. We thus evaluated the effect of COA6 depletion on mitochondrial respiration of COAD cells. By comparing the respiration profiles of siCtrl with siCOA6 cells, we observed reduced respiration in knockdown cells (Figs 5A and B and S5A and B). This finding is consistent with the recognized role of COA6 in facilitating complex IV biogenesis and activity. In addition, similar to FKBP4 depletion,

COA6 knockdown had no effect on glycolysis or glycolytic capacity of COAD cells (Fig S5C–F). Moreover, ectopic expression of MTS1-FKBP4 could not rescue the impaired respiration upon COA6 loss (Figs 5C and D and S5G and H). These data demonstrate that COA6 is indispensable for maintaining mitochondrial respiration in COAD cells. To further assess the impact of COA6 loss on 5-FU sensitivity of COAD cells, we examined the cell viabilities of HCT116 and SW480 cells upon treatment with siCtrl or siCOA6. The results showed that IC50 of 5-FU in COAD cells was dramatically reduced upon COA6 knockdown (Figs 5E and S6A). Similar to the results of respiration

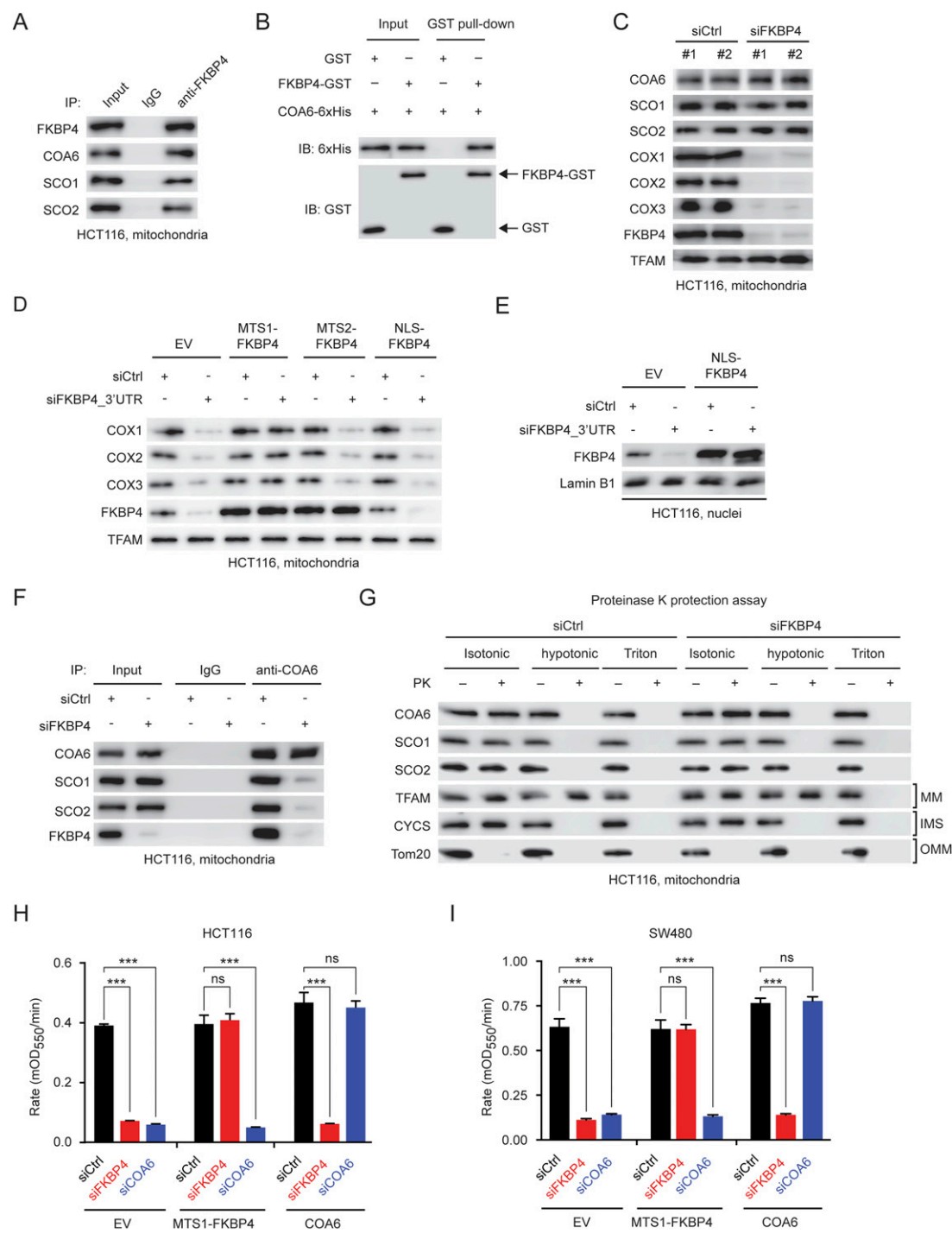

**Figure 4. FKBP4 controls COA6-regulated complex IV activity.**
**(A)** Immunoprecipitation of FKBP4 in mitochondria of HCT116 cells with anti-FKBP4 antibody, followed by immunoblotting with indicated antibodies. **(B)** GST pull-down analysis of interaction between FKBP4 and COA6 using recombinant GST-tagged FKBP4 and 6xHis-tagged COA6. **(C)** Western blot analysis of the indicated proteins in the mitochondrial fraction of HCT116 cells treated with two different siCtrl or siFKBP4. TFAM was used as the loading control. **(D, E)** Western blot analysis of the indicated proteins in the mitochondrial (D) and nuclear fraction (E) of HCT116 cells transduced with empty vector (EV), MTS1-FKBP4, MTS2-FKBP4, or NLS-FKBP4 in combination with siCtrl or siFKBP4_3′UTR. **(F)** Immunoprecipitation of COA6 in the mitochondrial fraction of siCtrl or siFKBP4 HCT116 cells, followed by immunoblotting with indicated antibodies. **(G)** Proteinase K protection assay of mitochondria isolated from siCtrl or siFKBP4-treated HCT116 cells. PK, proteinase K; MM, mitochondrial matrix; IMS, intermembrane space; OMM, outer mitochondrial membrane. TFAM (MM protein), CYCS (IMS protein), and Tom20 (OMM protein) were used as the control showing the extent of digestion. **(H, I)** Complex IV activity in HCT116 (H) and SW480 (I) cells transfected with an empty vector (EV), MTS1-FKBP4, or COA6 in combination with siCtrl, siFKBP4_3′UTR, or siCOA6_3′UTR treatment. ***$P < 0.001$; ns, not significant by two-tailed $t$ test. Data are presented as mean + SD. n = 6.

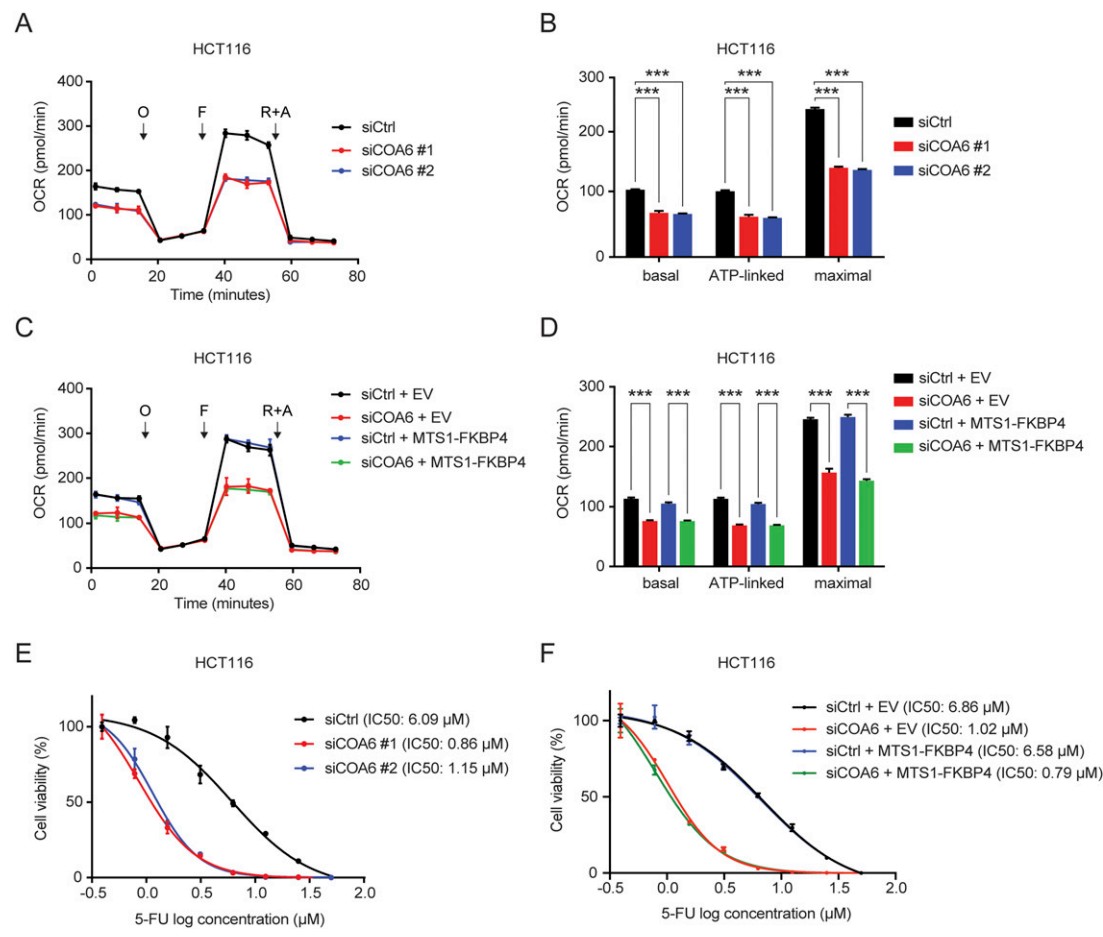

**Figure 5. COA6 modulates mitochondrial respiration and sensitivity of COAD cells to 5-FU.**
**(A, B)** Mitochondrial respiration profiles of HCT116 cells treated with siCtrl or two different siCOA6. O, oligomycin; F, carbonyl cyanide-4-(trifluoromethoxy)phenylhydrazone (FCCP); R+A, rotenone and antimycin A. **(B)** Basal, ATP-linked respiration, and maximal respiratory capacity were calculated. ***$P < 0.001$ by two-tailed $t$ test. **(C, D)** Profiles of mitochondrial respiration over time in HCT116 cells transfected with an empty vector (EV) or MTS1-FKBP4 in combination with the treatment of siCtrl or siRNA targeting 3′UTR of endogenous COA6 mRNA (siCOA6). **(D)** Basal, ATP-linked respiration, and maximal respiratory capacity were calculated. ***$P < 0.001$ by the two-tailed $t$ test. **(A, B, C, D)** Data are presented as mean ± SD. n = 6. **(E)** Representative dose–response curves of 5-FU treatment in HCT116 cells transfected with siCtrl or two different siRNA targeting COA6. **(F)** Representative dose–response curves of 5-FU treatment in HCT116 cells transfected with an empty vector (EV) or MTS1-FKBP4 in combination with the treatment of siCtrl or siRNA targeting 3′UTR of endogenous COA6 mRNA (siCOA6). **(E, F)** Twenty-four hours after siRNA transfection, cells were seeded into 96-well plate. After 24 h, cells were treated with different doses of 5-FU for 72 h, followed by cell viability analysis using SRB assay. IC50 values of 5-FU were indicated. Data are presented as mean ± SD. n = 3.

analysis, ectopic expression of MTS1-FKBP4 failed to reverse the decreased IC50 values upon COA6 loss (Figs 5F and S6B). Therefore, these findings indicate that COA6 is critical in sustaining mitochondrial respiration, thus affecting sensitivity of COAD cells to 5-FU.

### Inhibition of complex IV phenocopies the enhanced 5-FU sensitivity upon FKBP4 or COA6 depletion in COAD cells

To further assess the contribution of complex IV inhibition to 5-FU sensitivity, we used two well-recognized complex IV inhibitors, KCN (Khutornenko et al, 2010; Yamamoto et al, 2012; Zhang et al, 2017) and ADDA5 (Oliva et al, 2016; Remsik et al, 2020), to study their impact on 5-FU sensitivity in COAD cells. To this end, we determined the cell viability of COAD cells upon treatment with KCN or ADDA5 using SRB assay. We observed that either of these two inhibitors caused dramatic decline in IC50 values of 5-FU in COAD cells (Fig 6A–D). These findings demonstrate that the inhibition of complex IV

activity is sufficient to increase the sensitivity of COAD cells to 5-FU. Importantly, ectopic expression of MTS1-FKBP4 failed to reverse the complex IV inhibitor–triggered enhancement of 5-FU sensitivity (Fig 6A–D). These findings suggest that the role of IMS-localized FKBP4 in the regulation of 5-FU sensitivity is dependent on its manipulation of complex IV activity.

In summary, our data suggest a novel role of FKBP4 in the regulation of mitochondrial respiration via modulating COA6-mediated mitochondrial complex IV biogenesis and activity, thus controlling the sensitivity of COAD cells to 5-FU.

## Discussion

The COA6/SCO1/SCO2 complex is essential for the assembly and activity of complex IV, thereby controlling mitochondrial respiration (Pacheu-Grau et al, 2015, 2020; Stroud et al, 2015; Soma et al, 2019;

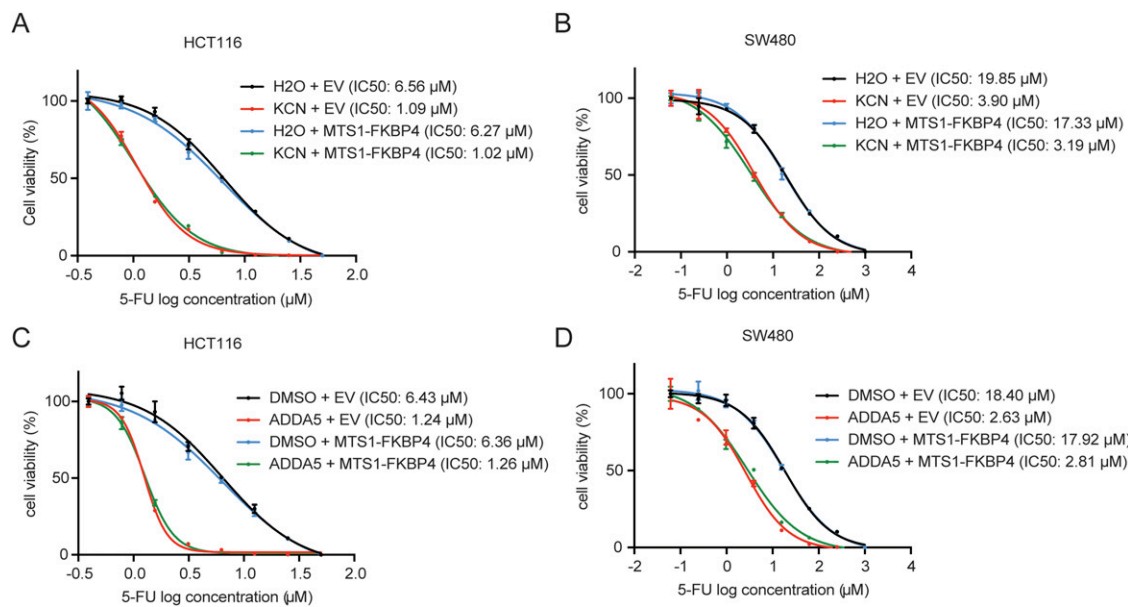

**Figure 6. Complex IV inhibition enhances sensitivity of COAD cells to 5-FU.**
**(A, B)** Representative dose–response curves of 5-FU treatment in HCT116 (A) and SW480 (B) cells transfected with an empty vector (EV) or MTS1-FKBP4 in combination with the treatment with H2O or 20 μM KCN (potassium cyanide). **(C, D)** Representative dose–response curves of 5-FU treatment in HCT116 (C) and SW480 (D) cells transfected with empty vector (EV) or MTS1-FKBP4 in combination with the treatment with DMSO or 10 μM ADDA5 (ADDA 5 hydrochloride). **(A, B, C, D)** Cells were treated with different doses of 5-FU together with 20 μM KCN or 10 μM ADDA5 for 72 h, followed by cell viability analysis using SRB assay. IC50 values of 5-FU were indicated. Data are presented as mean ± SD. n = 3.

Maghool et al, 2020; Swaminathan & Gohil, 2022). Mitochondrial dysfunction caused by COA6 loss-of-function mutation leads to several types of human mitochondrial diseases (Ghosh et al, 2014; Baertling et al, 2015; Pacheu-Grau et al, 2015). Nevertheless, being a mitochondrial respiration-dependent disease, COAD has not yet been linked to COA6-modulated mitochondrial function. Our study fills this gap by showing that COA6 depletion sensitized the COAD cells to 5-FU treatment via impairing complex IV activity.

The co-chaperone protein FKBP4 is characterized as a scaffold for sustaining the assembly of multi-protein complexes (Mange et al, 2019; Zong et al, 2021). In alignment with this, we showed that FKBP4 depletion led to rupture of the COA6/SCO1/SCO2 complex, suggesting a critical role of FKBP4 in maintaining the assembly of this complex. Thus, our study extends the knowledge on the establishment of the COA6/SCO1/SCO2 complex and has identified new important interacting partner of this functional complex.

Although FKBP4 has been shown to reside in both the cytoplasm and nucleus in various cell types (Thul et al, 2017), the subcellular compartmentalization of FKBP4 and its role in cancer cells has not yet been addressed. Our study investigated the subcellular distribution of FKBP4 and identified a significant population of FKBP4 in the mitochondria of COAD cells. Further assessment of submitochondrial localization, we found that FKBP4 resides both in IMS and MM. Importantly, function analysis identified that IMS-localized, instead of MM-localized or nucleus-localized FKBP4, is critical for mitochondrial respiration of COAD cells. Thus, our study, for the first time, dissects the compartmentalized activity of FKBP4 in COAD cells.

Most of the mitochondrial proteins are encoded by the nuclear genome, translated in the cytoplasm, and imported into mitochondria (Pfanner et al, 2019). These nuclear-transcribed mitochondrial proteins frequently contain MTSs within their N-terminal amino acid sequence (Wiedemann & Pfanner, 2017; Hansen & Herrmann, 2019). Although FKBP4 showed mitochondrial localization, no clear canonical MTS was detected in it. It might be that FKBP4 has unconventional targeting signal that cannot be identified by bioinformatic prediction, as seen in MOF, APE1, CREB, and p53 (Marchenko et al, 2000; Lee et al, 2005; Li et al, 2010; Chatterjee et al, 2016). Mitochondrial import of FKBP5, another member of the FKBP family, is known to be conferred by its TPR domain through a HSP90/HSP70–dependent mechanism (Gallo et al, 2011). Although FKBP4 also contains the TPR domain, its mitochondrial localization is not dictated by the TPR domain–mediated interaction with HSP90/HSP70. Thus, future work is needed to identify the special mitochondrial import machinery of FKBP4.

FKBP4 is known to regulate nuclear translocation and/or transcriptional activity of several cancer-related transcription factors and thus contributes to malignancy and progression of several types of human cancer (Chen et al, 2010; Mange et al, 2019; Liu & Gao, 2021; Zong et al, 2021). However, no study has yet addressed whether FKBP4 plays a role in conferring the sensitivity of COAD cells to chemotherapy agents. Our study for the first time fills this gap by showing that targeting FKBP4/COA6 enhances the 5-FU sensitivity in COAD cells via inhibiting the activity of mitochondrial complex IV. Mechanistically, we found that FKBP4 regulates COA6-mediated mitochondrial cytochrome c oxidase biogenesis. In addition to what we have shown in the present study, FKBP4 was shown to facilitate breast cancer proliferation via activating the PI3K/AKT

pathway (Mange et al, 2019). Accordingly, FKBP5 that acts antagonistically to the function of FKBP4 (Barent et al, 1998; Davies et al, 2002; Galigniana et al, 2010) was found to confer chemosensitization via negatively regulating AKT in several types of cancer cells (Pei et al, 2009; Li et al, 2011). Thus, it would be interesting to examine whether FKBP4 confers chemoresistance in COAD via modulating the PI3K/AKT signaling pathway in future work. In the present study, we showed that direct inhibition of complex IV using selective chemical inhibitors improves the response of COAD cells to 5-FU. Thus, our data shed light on potential advantages for combination treatment with 5-FU and complex IV inhibitors in anti-COAD therapy.

## Materials and Methods

### Antibodies, siRNAs, plasmids, and chemicals

Detailed information regarding the antibodies, siRNAs, plasmids, and chemicals used in this study is provided in Table S1.

### Cell culture, cloning, and transfection

HCT116 and SW480 colon cancer cells were obtained and authenticated from ATCC. HCT116 and SW480 cells were maintained in DMEM supplemented with 10% FBS, glutamine, and penicillin/streptomycin. Cells were tested as mycoplasma-free using the Universal Mycoplasma Detection Kit (30-1012K; ATCC). The MTS sequence from SMAC (referred to as MTS1, MAALKSWLSRSVTSF-FRYRQCLCVPVVANFKKRCFSELIRPWHKTVTIGFGVTLCAVPI) was fused to the N-terminus of FKBP4 by cloning to import the FKBP4 protein into the mitochondrial intermembrane space. The MTS sequence from COX4A (referred to as MTS2, MLATRVFSLVGKRAISTSVCVRAH) was fused to the N-terminus of FKBP4 by cloning to import the FKBP4 protein into the mitochondrial matrix. The nuclear localization sequence from c-Myc (referred to as NLS, PAAKRVKLD) was fused to the N-terminus of FKBP4 by cloning to import the FKBP4 protein into the nucleus. For ectopic expression of these FKBP4-derivatives, lentiviral plasmids were used to produce recombinant lentiviruses to infect HCT116 and SW480 cells. For generation of siRNA-mediated knockdown, cells were transfected with siCtrl, siFKBP4, or siCOA6 using Lipofectamine RNAiMAX Transfection Reagent (Thermo Fisher Scientific) according to the manufacturer's instructions, followed by harvesting the cells for the subsequent assay.

### Cell fractionation and mitochondrial isolation

Cell fractionation and mitochondrial isolation were performed with the Mitochondria Isolation Kit for Cultured Cells (Cat. no.: ab110170; Abcam) according to the manual. In detail, cells were frozen and thawed once in liquid nitrogen to weaken the cell membranes and then resuspended with Reagent A (supplied in the kit) and incubated for 10 min on ice followed by homogeniation with the Dounce homogenizer. Next, homogenate was clarified with centrifugation at 1,000$g$ for 10 min at 4°C. After centrifugation, supernatant was

saved as supernatant #1. The pellet was resuspended with Reagent B (supplied in the kit) and homogenized with the Dounce homogenizer. Homogenate was then clarified with centrifugation at 1,000$g$ for 10 min at 4°C. After centrifugation, supernatant was saved as supernatant #2. The pellet (nuclei) was lysed with RIPA lysis buffer (50 mM Tris, pH 7.4, 150 mM NaCl, 1% Nonidet P-40, 0.1% SDS, 0.5% sodium deoxycholate, and cOmplete protease inhibitor Cocktail [Sigma-Aldrich]) to get the nuclear fraction. Supernatant #1 and #2 were combined, mixed thoroughly, and centrifuged at 12,000$g$ for 15 min at 4°C. After centrifugation, supernatant was collected (cytosolic fraction). The pellet (mitochondria) was lysed with RIPA lysis buffer to get the mitochondrial fraction.

### Western blot analysis

Equal amounts of proteins from each sample were subjected to SDS–PAGE separation, followed by transferring to a PVDF membrane. Membranes were then blocked with blocking buffer (PBS containing 0.1% Tween 20 and 5% wt/vol nonfat dry milk) for 1 h at room temperature, followed by incubation with indicated primary antibody at 4°C overnight. The next day, membranes were washed three times with washing buffer (PBS containing 0.1% Tween 20), followed by incubation with HRP-conjugated secondary antibody for 1 h at room temperature. Next, membranes were washed three times with washing buffer and then subjected to treatment with ECL solution and chemiluminescent detection.

### Proteinase K protection assay

Mitochondria were freshly isolated with the Mitochondria Isolation Kit for Cultured Cells (Cat. no.: ab110170; Abcam) as described above and then resuspended with each of the three following buffers: (1) isotonic buffer (10 mM MOPS-KOH, pH 7.2, 250 mM sucrose and 1 mM EDTA), (2) hypotonic buffer (10 mM MOPS-KOH, pH 7.2, and 1 mM EDTA) and (3) Triton buffer (10 mM Tris–HCl, pH 7.4, 1 mM EDTA, and 1% Triton X-100), followed by treatment with 50 $\mu$g/ml proteinase K for 15 min on ice. Reaction was then terminated with 20 mM PMSF. Proteins from each reaction were precipitated with 10% trichloroacetic acid (TCA), followed by washing with acetone, resuspension with 1× SDS loading buffer, and heating at 95°C for 10 min. Finally, the samples were subjected to Western blot analysis.

### Co-immunoprecipitation

Mitochondria of HCT116 cells were isolated with the Mitochondria Isolation Kit for Cultured Cells (Cat. no.: ab110170; Abcam) according to the manual as described above. Freshly isolated mitochondria were then lysed in co-IP buffer (50 mM Tris, pH 7.4, 150 mM NaCl, 2 mM EDTA, 1% NP-40, and cOmplete protease inhibitor Cocktail [Sigma-Aldrich]). Extracts were clarified by centrifugation. The lysates were then incubated with the indicated antibodies overnight at 4°C followed by incubation with Protein-A/G-Sepharose beads (GE Healthcare) for 2 h. Immunoprecipitates were washed five times in co-IP buffer, dissolved in 5× SDS–PAGE sample buffer, and subjected to standard Western blot analysis.

## Mitochondrial respiration measurement

Mitochondrial respiration was assessed by using the Seahorse XF Cell Mito Stress Test Kit (Agilent) on the Seahorse XFe 96 analyzer (Agilent) based on manufacturer's instruction. In brief, cells were treated with siCtrl or indicated siRNA for 2 d, followed by trypsinization and cell counting. Next, 20,000 cells were seeded in XF 96 cell culture microplates and allowed to attach overnight. The next day, cell culture medium was replaced with Seahorse XF DMEM medium (Agilent). After 1 h incubation at 37°C in a $CO_2$-free incubator, cells were subjected to Seahorse XF Cell Mito Stress Test assay for measuring the oxygen consumption rate values with the sequential addition of oligomycin (1 $\mu$M), FCCP (1 $\mu$M), and a mix of rotenone (2 $\mu$M) and antimycin A (2 $\mu$M). The basal, ATP-linked, and maximal respiration were calculated according to the manual. After the run, assay media was removed, and cells were lysed for Bradford protein assay to measure protein concentration. The data were normalized with protein concentration.

## Determination of glycolytic function

Glycolytic levels and capacities of colon cancer cells were determined by using the Seahorse XF Glycolysis Stress Test Kit (Agilent) on the Seahorse XFe 96 analyzer (Agilent) based on manufacturer's instruction. In brief, 15,000 cells were seeded in XF 96 cell culture microplates, followed by overnight incubation at 37°C, 5% $CO_2$ to allow the adhesion of the cells. The next day, cell culture medium was replaced with Seahorse XF DMEM medium (Agilent). After incubation at 37°C in a $CO_2$-free incubator for 45 min, cells were subjected to Seahorse XF Glycolysis Stress Test assay for measuring the extracellular acidification rate (ECAR) values with the sequential addition of glucose (10 mM), oligomycin (1 $\mu$M), and 2-deoxyglucose (50 mM). The glycolysis and glycolytic capacity were calculated according to the manual. After the assay was finished, assay media was removed, and cells were lysed and then proceeded to protein concentration measurement using Bradford protein assay. The data were normalized with protein concentration.

## Sulforhodamine B (SRB) assay

Cells were seeded into 96-well plates at a density of 3,000 cells per well. The next day, cells were treated with various concentrations of 5-FU as indicated in. After 72 h of drug exposure, SRB assay was performed with the Sulforhodamine B Assay Kit (Abcam) according to the manual. In brief, cells were fixed with Fixation Solution for 1 h at 4°C, followed by removal of solution and three times gently washing with $ddH_2O$. Next, the cells were subjected to staining with SRB solution for 15 min at room temperature in the dark. SRB solution was then removed, and cells were washed with Washing Solution for four times. The plate was then air-dried before the addition of Solubilization Solution and incubation at room temperature for 10 min. The absorbance was measured at 565 nm. All the solutions used in the assay were supplied with the kit. The half-maximal inhibitory concentration (IC50) values were determined by fitting a 4-parameter logistic (4PL) sigmoidal dose–response curve to the data.

## Measurements of complex IV activity

Mitochondria of COAD cells were extracted as described above. Complex IV activity of freshly isolated mitochondria was measured using a Complex IV Human Enzyme Activity Microplate Assay Kit (ab109909; Abcam) according to the manufacturer's protocol.

## Data collection

COAD transcriptome mRNA-seq data and corresponding clinical dataset were downloaded from TCGA (https://cancergenome.nih.gov, TCGA-COAD).

## Statistical analysis

Statistical analyses for the comparison of expression levels of FKBP4 and COA6 between normal colon tissues and COAD specimens were performed using the Wilcoxon rank-sum test. Results from quantitative analyses of other experiments are presented as mean + SD. Statistical analysis was performed using the $t$ test with two-tailed distribution. Sample sizes are indicated where appropriate and were determined based on previous experience with analogous experiments. Statistical significance is presented as follows: $*P < 0.05$, $**P < 0.01$, $***P < 0.001$.

# Data Availability Statement

A publicly available dataset was analyzed in this study. These data can be found here: TCGA (https://portal.gdc.cancer.gov/).

# Supplementary Information

# Acknowledgements

We thank the TCGA Research Network for providing its platforms and valuable data sets. This present study was funded by the Shandong Province Major Science and Technology Innovation Project (Grant no. 2019JZZY011008).

## Author Contributions

Z Zhu: conceptualization, validation, investigation, methodology, and writing—original draft, review, and editing.
Q Hou: conceptualization, validation, investigation, methodology, and writing—review and editing.
B Wang: conceptualization and investigation.
C Li: conceptualization and validation.
L Liu: conceptualization, investigation, and methodology.
W Gong: validation, investigation, and methodology.
J Chai: conceptualization, resources, supervision, funding acquisition, validation, project administration, and writing—review and editing.

H Guo: conceptualization, resources, supervision, funding acquisition, project administration, and writing—review and editing.

Y Jia: conceptualization, resources, formal analysis, supervision, validation, investigation, methodology, project administration, and writing—review and editing.

## Conflict of Interest Statement

The authors declare that the research was conducted in the absence of any commercial or financial relationships that could be construed as a potential conflict of interest.

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
