## [Reviewer comments · Life Science Alliance]

Life Science Alliance

FKBP4 regulates 5-fluorouracil sensitivity in colon cancer by controlling mitochondrial respiration

Zhenyu Zhu, Qingsheng Hou, Bishi Wang, Changhao Li, Luguang Liu, Weipeng Gong, Jie Chai, Hongliang Guo, Yanhan Jia
DOI: <https://doi.org/10.26508/lsa.202201413>

Corresponding author(s): Dr. Yanhan Jia (Sichuan Cancer Hospital & Institute, Sichuan Cancer Center, School of Medicine, University of Electronic Science and Technology of China, Chengdu, China)

Review Timeline:	Submission Date:	2022-02-16
	Editorial Decision:	2022-05-23
	Revision Received:	2022-07-23
	Editorial Decision:	2022-07-31
	Revision Received:	2022-08-05
	Accepted:	2022-08-09

Scientific Editor: Novella Guidi

Transaction Report:

May 23, 2022

Re: Life Science Alliance manuscript #LSA-2022-01413-T

Dr. Yanhan Jia
Sichuan Cancer Hospital & Institute, Sichuan Cancer Center, School of Medicine, University of Electronic Science and Technology of China
No. 55 Renmin South Road
Chengdu, Sichuan 610041
China

Dear Dr. Jia,

Thank you for submitting your manuscript entitled "FKBP4 regulates 5-fluorouracil sensitivity in colon cancer by controlling COA6-mediated mitochondrial respiration" to Life Science Alliance. The manuscript was assessed by expert reviewers, whose comments are appended to this letter. We invite you to submit a revised manuscript addressing the Reviewer comments.

Thank you for this interesting contribution to Life Science Alliance. We are looking forward to receiving your revised manuscript.

Sincerely,

B. MANUSCRIPT ORGANIZATION AND FORMATTING:

Reviewer #1 (Comments to the Authors (Required)):

Comments to the Authors:

FKBP4 (FKBP52) is a member of the immunophilin family which are involved in numerous cellular functions, such as protein folding and stability, kinase activity or protein trafficking. FKBP4 is overexpressed in cancers including breast and prostate cancers with a possible role of this protein in tumorigenesis and tumor progression through the regulation of the activity of steroid hormone receptor or cell signaling pathways. In this manuscript, the authors describe that FKBP4 is overexpressed in colon adenocarcinoma and is essential for the maintenance of mitochondrial respiration and for the sensitivity of COAD cells to 5-FU. Although the question raised by the authors is new and interesting, the data do not support all their conclusion. Especially, lack of the mechanism is a concern of this study.

There are open problems to be addressed prior to publication, as indicated below:

1-In Figure 1, authors show clearly the overexpression of FKBP4 in COAD cells and its localization in both mitochondrial intermembrane space and matrix. Page 3, authors could not observe any canonical mitochondrial targeting sequences in FKBP4. However, Gallo et al. (J. Biol. Chem, 10.1074/jbc.M111.256610) showed that FKBP5 (FKBP51), an antagonist of FKBP4 sharing more than 60% identity, was found also in mitochondria and can translocate to the nucleus. Moreover, they showed that the tetratricopeptide repeat motif (TPR), also present in FKBP4, is required for its mitochondrial localization through a HSP90/HSP70-dependent mechanism. Some outer mitochondrial membrane transporter, such as Tom70 or Tom20, contain also TPR domain providing a possible docking site. Is it the same mechanism for FKBP4? What could be the impact of mutated TPR of FKBP4 in the COAD cells? Please, cite the Gallo et al. study.

Figure 1 (or supplementary figure) should include a western blot with cells treated with siCtrl and both siFKBP4-1/2.

2- In Figure 3, authors show that FKBP4 in the mitochondrial intermembrane space enhanced the sensitivity of COAD cells to 5-FU. It seems clear to me that ectopic expression of MTS1-FKBP4 reverse the increased 5-Fu sensitivity. However, I wonder about the lack of effect with ectopic expression of MTS2-FKBP4. Indeed, 5-FU resistance can be attributed not only to metabolic switch toward oxidative phosphorylation but also to other mechanisms such as the acquisition of quiescent state, aberrant activation of different cell survival signaling pathways, and resistance to DNA damage. Considering the pleiotropic effect of FKBP4 on cell signaling pathways, cell cycle or proliferation, have the authors check the impact of siFKBP4 and/or ectopic expression of MTS1-, MST2- and NLS-FKBP4 on cell growth, proliferation, cell cycle or apoptosis rate, which could modify the IC50 determination? Please explain what is the reference (100%)? Control cells not treated with 5-FU?

3- Figure 4A: Page 6: at this stage, the authors do not show data that ensures that FKBP4 interact directly to COA6. Co-IP show only that these proteins are in the same proximal environment. Are Cox-1, Cox-2 and Cox-3 detected in the same complex?

Figure 4B: How could you explain the "downregulation" of the key subunits of complex IV (transcriptional level? Translational level?) and what is the role of FKBP4 in this downregulation? As many subunits of cytochrome oxidase and translational activator proteins are encoded in the nucleus and imported into mitochondria from the cytosol, FKBP4 could be involved in this intracellular trafficking or as co-chaperone of the translational complex. What is the effect of ectopic expression of MST1- and MST2-FKBP4 on the Cox expression?

Figure 4C: Authors clearly show that FKBP4 is indispensable for the assembly of COA6/SCO1/SCO2 complex in mitochondria. As these proteins are encoded in the nucleus and imported into mitochondria, why not look more in depth their localization (in MM, IMS and OMM) using proteinase K protection assay in siCtrl and siFKBP4 cells? Are COA6 and SCO1/2 in the same compartment in siFKBP4 cells?

4- Figure 7 and text: Concerning the expression of FKBP4 and COA6 in CIAD patients, please indicate the ID of the dataset used, and more details on the number of patients (normal vs COAD). Why used Wilcoxon rank-sum test? Without any covariate

analysis, why used Univariate Cox regression rather than Kaplan-Meier and log-rank-test for survival analysis?

5- Discussion:

Page 9, 3rd paragraph, discuss and cite Gallo et al. (J. Biol. Chem, 10.1074/jbc.M111.256610).

Page 9, last paragraph: As co-chaperones, FKBP4 and FKBP5 are known to be associated and to exhibit antagonist functions into the regulation of the steroid receptors, the transcriptional activity of NF- κ B or the PI3K/Akt signaling pathway. For FKBP5, it has been reported that this protein is a negative regulator of cell growth by inhibiting the PI3K/Akt pathway which can influence the response of chemotherapy (Pei et al., Cancer Cell, 10.1016/j.ccr.2009.07.016; Li et al., Br J Cancer, 10.1038/sj.bjc.6606014). As FKBP4 is a positive regulator of the same pathway (Mange et al., Theranostics), please discuss these additional hypotheses.

Minor comments:

Page 10: Please change "FKBP5/COA6 axis" by "FKBP4/COA6".

Reviewer #3 (Comments to the Authors (Required)):

Role of FKBP4 has been shown in some cancer types and the authors have studied COAD here which is novel.

General:

It would be incorrect to say that cancer cells rely on glycolysis. There are numerous studies showing just how essential mitochondrial metabolism is for cancer progression. It would be fair to say eg that COAD has more mitochondrial metabolism as compared to a different cancer type.

Experiments are well-designed and hypothesis based however the methods and results are poorly explained.

What are the effects of FKBP4 on proliferation, apoptosis, migration, and invasion in COAD?

1D-G: How has the data been normalized here? Did the authors measure cell death due to FKBP4 KD? If there is cell death which makes sense if you are knocking down a protein essential for cancer cells, then how were these assays normalized? What is the level of KD achieved by these siRNA's? Were cells treated with siRNA before or after plating the cells for Seahorse analysis.

Fig 2: Are the authors knocking down FKBP4 first and then expressing the IMS-localized version of this protein? This is very difficult to understand.

Point-by-point response to the reviewers' comments

We would like to express our sincere gratitude to Editorial Office and all the reviewers for their appreciation of our work and especially for their thoughtful and constructive comments, which helped us to improve the quality of our manuscript considerably and to clarify a number of important points in the following point-by-point response. All major changes in the revised manuscript are highlighted in yellow.

The reviewer's comments are in italics.

Reviewer #1:

1-In Figure 1, authors show clearly the overexpression of FKBP4 in COAD cells and its localization in both mitochondrial intermembrane space and matrix. Page 3, authors could not observe any canonical mitochondrial targeting sequences in FKBP4. However, Gallo et al. (J. Biol. Chem, 10.1074/jbc.M111.256610) showed that FKBP5 (FKBP51), an antagonist of FKBP4 sharing more than 60% identity, was found also in mitochondria and can translocate to the nucleus. Moreover, they showed that the tetratricopeptide repeat motif (TPR), also present in FKBP4, is required for its mitochondrial localization through a HSP90/HSP70-dependent mechanism. Some outer mitochondrial membrane transporter, such as Tom70 or Tom20, contain also TPR domain providing a possible docking site. Is it the same mechanism for FKBP4? What could be the impact of mutated TPR of FKBP4 in the COAD cells? Please, cite the Gallo et al. study.

Response: We thank the reviewer for this comment. During the revision of the manuscript, we generated the COAD cells harboring the K354A mutation in the TPR domain of FKBP4 (FKBP4-K354A), that has been widely shown to abolish its interaction with HSP90. In consistence with the previously published data, we observed that FKBP4-K354A mutation abrogated the FKBP4/HSP90 and FKBP4/HSP70 interaction (Figure S1B), but did not affect the mitochondrial localization of FKBP4 (Figure S1C). These finding indicates that, unlike FKBP5, the mitochondrial localization of FKBP4 is not controlled by its TPR domain-mediated interaction with HSP90 /HSP70.

Figure 1 (or supplementary figure) should include a western blot with cells treated with siCtrl and both siFKBP4-1/2.

Response: We thank the reviewer for this comment. In the revised version of the manuscript, we added the western blot data in Figure 1E and F.

2- In Figure 3, authors show that FKBP4 in the mitochondrial intermembrane space enhanced the sensitivity of COAD cells to 5-FU. It seems clear to me that ectopic expression of MTS1-FKBP4 reverse the increased 5-Fu sensitivity. However, I wonder about the lack of effect with ectopic expression of MTS2-FKBP4. Indeed, 5-FU resistance can be attributed not only to metabolic switch toward oxidative phosphorylation but also to other mechanisms such as the acquisition of quiescent state, aberrant activation of different cell survival signaling pathways, and resistance to DNA damage. Considering the pleiotropic effect of FKBP4 on cell signaling pathways, cell cycle or proliferation, have the authors check the impact of siFKBP4 and/or ectopic expression of MTS1-, MST2- and NLS-FKBP4 on cell growth, proliferation, cell cycle or apoptosis rate, which could modify the IC50 determination? Please explain what is the reference (100%)? Control cells not treated with 5-FU?

Response: We thank the reviewer for this comment. During the revision of the manuscript, we found that FKBP4 knockdown does not influence the cell growth, proliferation, apoptosis rate, migration, or invasion of COAD cells (Figure S4). Regarding the dose-response curves of 5-FU treatment in COAD cells, the cell viability was determined by SRB assay. The reference (100%) is the absorbance value (OD₅₆₅) of the cells treated with the lowest concentration of the 5-FU. For example, in the case of Figure 3A, siCtrl #1, siCtrl #2, siFKBP4 #1, siFKB4 #2 HCT116 cells were treated with different concentration of 5-FU (50, 25, 12.5, 6.3, 3.1, 1.6, 0.8, 0.4 μ M) in order to make the dose-response curve, the 100% of each group was the OD₅₆₅ upon 0.4 μ M 5-FU treatment.

3- Figure 4A: Page 6: at this stage, the authors do not show data that ensures that FKBP4 interact directly to COA6. Co-IP show only that these proteins are in the same proximal environment. Are Cox-1, Cox-2 and Cox-3 detected in the same complex?

Response: We thank the reviewer for this comment. During the revision of the manuscript, we performed the GST pull-down analysis and confirmed that FKBP4 directly interacts with COA6 (Figure 4B). COA6 is widely shown to dictate the translation of mtDNA-encoded COXs (Aich et al., 2018; Pacheu-Grau et al., 2015; Stroud et al., 2015). Pacheu-Grau *et al.* has identified that COA6 interacts only transiently with COX2 and it only happens early after COX2 synthesis (Pacheu-Grau et al., 2015). Thus, the COX1, COX2, and COX3 are not likely to be in the same complex. Actually, in consistence with Pacheu-Grau *et al.*'s study, we also failed to detect any interaction of FKBP4 with COX1, COX2, or COX3.

Figure 4B: How could you explain the "downregulation" of the key subunits of complex IV (transcriptional level? Translational level?) and what is the role of FKBP4 in this downregulation? As many subunits of cytochrome oxidase and translational activator proteins are encoded in the nucleus and imported into mitochondria from the cytosol, FKBP4 could be involved in this intracellular trafficking or as co-chaperone of the translational complex. What is the effect of ectopic expression of MTS1- and or MTS2-FKBP4 on the Cox expression?

Response: We thank the reviewer for this comment. COA6/SCO1/SCO2 complex has been widely shown to guide the translation of the mtDNA-encoded COXs (Aich et al., 2018; Pacheu-Grau et al., 2015; Stroud et al., 2015). Previous studies have also shown that loss of COA6 led to decrease in protein levels of mitochondrial COXs including COX1, COX2, and COX3 (Pacheu-Grau et al., 2015; Stroud et al., 2015). In our study, we found that loss of FKBP4 resulted in rupture of COA6/SCO1/SCO2 complex, thus leading to decreased levels of COXs. During the revision of the manuscript, we assessed the impact of MTS1-, MTS2-, and NLS-FKBP4 on the expression of COXs and found that only MTS1-FKBP4 could rescue the decreased levels of COX1, COX2, and COX3 upon FKBP4 depletion, indicating that only IMS-localized FKBP4 is essential for maintaining the expression of COXs (Figure 4D and E).

Figure 4C: Authors clearly show that FKBP4 is indispensable for the assembly of COA6/SCO1/SCO2 complex in mitochondria. As these proteins are encoded in the nucleus and imported into mitochondria, why not look more in depth their localization (in MM, IMS and OMM) using proteinase K protection assay in siCtrl and siFKBP4 cells? Are COA6 and SCO1/2 in the same compartment in siFKBP4 cells?

Response: We thank the reviewer for this comment. During the revision of the manuscript, we assessed the submitochondrial localization of COA6, SCO1, and SCO2 in both siCtrl and siFKBP4 cells. We found that all these three proteins are located in the same compartment (intermembrane space) in both siCtrl and siFKBP4 cells (Figure 4G). FKBP4 knockdown did not influence their submitochondrial localization (Figure 4G).

4- Figure 7 and text: Concerning the expression of FKBP4 and COA6 in COAD patients, please indicate the ID of the dataset used, and more details on the number of patients (normal vs COAD). Why used Wilcoxon rank-sum test? Without any covariate analysis, why

used Univariate Cox regression rather than Kaplan-Meier and log-rank-test for survival analysis?

Response: We thank the reviewer for this comment. In the revised manuscript, we have added the ID and the number of samples in both figure and figure legend.

The 'Wilcoxon Rank Sum test' (also called the 'Mann-Whitney test') is a non-parametric test that does not assume known distributions. It is widely used to compare the gene expression levels between two different sample groups (Li et al., 2022; Liu et al., 2021; Ruan et al., 2022; Zhang et al., 2022). Since the gene expression data of the TCGA are normally distributed (tested by Shapiro-Wilk Test), we also tried Student's t-test to do the statistical analysis and the p value (normal vs tumor) are less than 1×10^{-16} that is also significant.

During the revision of the manuscript, we have used the Kaplan-Meier and log-rank-test for the survival analysis. However, no significant correlation between FKBP4 or COA6 expression and prognosis/survival of 5-FU treated COAD patients was observed. It could be explained by the compartmentalized cellular distribution of FKBP4. In our study, we observed that FKBP4 is located in mitochondrial intermembrane space (IMS), mitochondrial matrix (MM) and nucleus (Figure 1B and C). However, only IMS-localized FKBP4 is important for regulating the 5-FU resistance of COAD cells (Figure 3). Thus, the current available online datasets, including TCGA, containing only the information of whole cellular FKBP4 expression level are not suitable to be used to investigate the prognostic significance of FKBP4 in 5-FU treated COAD. Therefore, considerable future work is needed to specifically assess the level of IMS-localized FKBP4 (not MM-localized or nuclear FKBP4) in COAD specimens as well as its potential link with COAD prognosis. Due to the above-mentioned reason, in the revised manuscript, we have removed the Figure 7B and 7C, and moved Figure 7A to Figure 1A for showing FKBP4 expression in human samples.

5- Discussion:

Page 9, 3rd paragraph, discuss and cite Gallo et al. (J. Biol. Chem, 10.1074/jbc.M111.256610).

Response: We thank the reviewer for this comment. In the revised manuscript, we have added the related discussion and citation according to the reviewer's comment.

Page 9, last paragraph: As co-chaperones, FKBP4 and FKBP5 are known to be associated and to exhibit antagonist functions into the regulation of the steroid receptors, the

transcriptional activity of NF- κ B or the PI3K/Akt signaling pathway. For FKBP5, it has been reported that this protein is a negative regulator of cell growth by inhibiting the PI3K/Akt pathway which can influence the response of chemotherapy (Pei et al., Cancer Cell, 10.1016/j.ccr.2009.07.016; Li et al., Br J Cancer, 10.1038/sj.bjc.6606014). As FKBP4 is a positive regulator of the same pathway (Mange et al., Theranostics), please discuss these additional hypotheses.

Response: We thank the reviewer for this comment. In the revised version of manuscript, we have added the related discussion following the reviewer's comment.

Minor comments:

Page 10: Please change "FKBP5/COA6 axis" by "FKBP4/COA6".

Response: We thank the reviewer for this comment and apologize for the mistake. In the revised manuscript, we have implemented the correction according to the reviewer's comment.

Reviewer #3:

It would be incorrect to say that cancer cells rely on glycolysis. There are numerous studies showing just how essential mitochondrial metabolism is for cancer progression. It would be fair to say eg that COAD has more mitochondrial metabolism as compared to a different cancer type.

Response: We thank the reviewer for this comment. In the revised manuscript, we have rephrased the related parts according to the reviewer's comment.

Experiments are well-designed and hypothesis based however the methods and results are poorly explained.

Response: We thank the reviewer for this comment. In the revised manuscript, we have rewritten the related parts with clearer explanations according to the reviewer's comment.

What are the effects of FKBP4 on proliferation, apoptosis, migration, and invasion in COAD?

Response: We thank the reviewer for this comment. During the revision of the manuscript, we found that FKBP4 depletion has no effect on proliferation, apoptosis, migration, or invasion of COAD cells (Figure S4).

1D-G: How has the data been normalized here? Did the authors measure cell death due to FKBP4 KD? If there is cell death which makes sense if you are knocking down a protein essential for cancer cells, then how were these assays normalized? What is the level of KD achieved by these siRNA's? Were cells treated with siRNA before or after plating the cells for Seahorse analysis.

Response: We thank the reviewer for this comment. As we have described it in the materials and methods section, after the assay has finished, assay media was removed and cells were lysed and proceeded to Bradford protein assay for measuring the protein concentration. The data were normalized with the protein concentration. We also observed that FKBP4 knockdown does not affect the growth and proliferation of COAD cells, and the protein concentrations of siCtrl and siFKBP4-treated COAD cells are quite equal, indicating an equal number of cells for the assay. In the revised version of the manuscript, we have added the immunoblotting data showing the knockdown efficiency of the siRNAs (Figure 1E and F). For the Seahorse analysis, cells were firstly treated with siCtrl or siFKBP4 for two days, followed by trypsinization, cell counting, and seeding into the assay plate for the Seahorse analysis.

Fig 2: Are the authors knocking down FKBP4 first and then expressing the IMS-localized version of this protein? This is very difficult to understand.

Response: We thank the reviewer for this comment and apologize for the confusion. For the rescue experiments, we first generated the MTS1-, MTS2-, or NLS-FKBP4 overexpressing cell lines via lentiviral transduction. Afterwards, the cells were treated with siCtrl or siFKBP4 for two days prior to subsequent assays.

References

Aich, A., Wang, C., Chowdhury, A., Ronsor, C., Pacheu-Grau, D., Richter-Dennerlein, R., Dennerlein, S., and Rehling, P. (2018). COX16 promotes COX2 metallation and assembly during respiratory complex IV biogenesis. *Elife* 7.

Li, C.H., Haider, S., and Boutros, P.C. (2022). Age influences on the molecular presentation of tumours. *Nat Commun* 13, 208.

Liu, J., Wang, X., Sun, J., Chen, Y., Li, J., Huang, J., Du, H., Gan, L., Qiu, Z., Li, H., *et al.* (2021). The Novel Methylation Biomarker NPY5R Sensitizes Breast Cancer Cells to Chemotherapy. *Front Cell Dev Biol* 9, 798221.

Pacheu-Grau, D., Bareth, B., Dudek, J., Juris, L., Vogtle, F.N., Wissel, M., Leary, S.C., Dennerlein, S., Rehling, P., and Deckers, M. (2015). Cooperation between COA6 and SCO2 in COX2 maturation during cytochrome c oxidase assembly links two mitochondrial cardiomyopathies. *Cell Metab* 21, 823-833.

Ruan, T., Wan, J., Song, Q., Chen, P., and Li, X. (2022). Identification of a Novel Epithelial-Mesenchymal Transition-Related Gene Signature for Endometrial Carcinoma Prognosis. *Genes (Basel)* 13.

Stroud, D.A., Maher, M.J., Lindau, C., Vogtle, F.N., Frazier, A.E., Surgenor, E., Mountford, H., Singh, A.P., Bonas, M., Oeljeklaus, S., *et al.* (2015). COA6 is a mitochondrial complex IV assembly factor critical for biogenesis of mtDNA-encoded COX2. *Hum Mol Genet* 24, 5404-5415.

Zhang, Y., Ma, Z., Li, C., Wang, C., Jiang, W., Chang, J., Han, S., Lu, Z., Shao, Z., Wang, Y., *et al.* (2022). The genomic landscape of cholangiocarcinoma reveals the disruption of post-transcriptional modifiers. *Nat Commun* 13, 3061.

July 31, 2022

RE: Life Science Alliance Manuscript #LSA-2022-01413-TR

Dr. Yanhan Jia

Sichuan Cancer Hospital & Institute, Sichuan Cancer Center, School of Medicine, University of Electronic Science and Technology of China, Chengdu, China
No. 55 Renmin South Road
Chengdu, Sichuan 610041
China

Dear Dr. Jia,

Thank you for submitting your revised manuscript entitled "FKBP4 regulates 5-fluorouracil sensitivity in colon cancer by controlling mitochondrial respiration". We would be happy to publish your paper in Life Science Alliance pending final revisions necessary to meet our formatting guidelines.

- Please make sure that here is a Title page of your ms file included within the Manuscript PDF file
- please make sure the author order in your manuscript and our system match
- please add a Summary Blurb/Alternate Abstract in our system
- please add a callout for Figure 6 to your main manuscript text (specific callouts A, B, C, D missing)
- please add your main, supplementary figure, and table legends to the main manuscript text after the references section;
- please add the Twitter handle of your host institute/organization as well as your own or/and one of the authors in our system

A. FINAL FILES:

B. MANUSCRIPT ORGANIZATION AND FORMATTING:

Sincerely,

Reviewer #1 (Comments to the Authors (Required)):

No particularly comments. The authors have responded to the various comments and completed their manuscript accordingly. The manuscript can be accepted for publication.

Reviewer #3 (Comments to the Authors (Required)):

The authors have answered my queries.

August 9, 2022

RE: Life Science Alliance Manuscript #LSA-2022-01413-TRR

Dr. Yanhan Jia
Sichuan Cancer Hospital & Institute, Sichuan Cancer Center, School of Medicine, University of Electronic Science and Technology of China, Chengdu, China
No. 55 Renmin South Road
Chengdu, Sichuan 610041
China

Dear Dr. Jia,

Thank you for submitting your Research Article entitled "FKBP4 regulates 5-fluorouracil sensitivity in colon cancer by controlling mitochondrial respiration". It is a pleasure to let you know that your manuscript is now accepted for publication in Life Science Alliance. Congratulations on this interesting work.

DISTRIBUTION OF MATERIALS:

Again, congratulations on a very nice paper. I hope you found the review process to be constructive and are pleased with how the manuscript was handled editorially. We look forward to future exciting submissions from your lab.

Sincerely,
